# Effects of Storage Method on the Quality of Processed Sea Cucumbers (*Apostichopus japonicus*)

**DOI:** 10.3390/foods11244098

**Published:** 2022-12-19

**Authors:** Shuang Li, Yan Zhou, Liming Sun, Yanjie Wang, Shuang Song, Chunqing Ai, Jingfeng Yang

**Affiliations:** Collaborative Innovation Center of Seafood Deep Processing, National Engineering Research Center of Seafood, School of Food Science and Technology, Dalian Polytechnic University, Dalian 116034, China

**Keywords:** sea cucumber, oxygenation, texture, ascorbic acid, protein degradation

## Abstract

This research aimed to establish an effective storage method to maintain the quality of processed sea cucumbers. In this study, sea cucumbers were stored by various methods including the storage of live sea cucumbers (seawater treatment, oxygen treatment, and ascorbic acid treatment) and the storage of dead sea cucumbers (frozen treatment). The sea cucumber quality was monitored after storage and boiling. The weightlessness rate and WHC of the frozen group increased to 86.96% ± 0.83% and 93.29% ± 0.32%, respectively. Frozen sea cucumbers shrunk with the meat’s textural properties deteriorated. During the live sea cucumber storage, the tissue protein degraded from day 3 to day 7 which led to the promotion of TVB-N. Among these, the oxygen group showed the smallest TVB-N increase from day 0 (3.78 ± 0.60 mg 100 g^−1^) to day 7 (10.40 ± 0.12 mg 100 g^−1^). The oxygen group exhibited the most moderate change in weightlessness rate (4.24% ± 0.45%) and the most moderate texture parameters decline, such as the hardness of 32.52%, chewiness of 78.98 ± 5.10 N, and adhesion of 0.84 ± 0.00. The oxygen method showed the best condition of sea cucumber after 5 days of storage.

## 1. Introduction

The sea cucumber is an invertebrate marine creature with high nutritional value. It is highly appreciated by consumers in China, Japan, and other Asian countries [1]. In recent years, with the increasing demand for sea cucumber and its products, the aquaculture industry has rapidly grown. The farmed sea cucumber output of 2021 reached 222,700 mt in China [2]. 

However, fresh sea cucumber is easy to autolyze and difficult to be transported and stored for a long time [3]. Thus, the sea cucumber storage and transportation method after harvest is a big challenge for the industry and it has a great influence on the processed sea cucumber quality. As a result, more than 80% of fresh sea cucumbers harvested around the world are sold by means of post-processing transportation [4,5]. Most sea cucumbers available for purchase are dehydrated by conventional techniques, including salting, repeated boiling, and exposure to solar radiation [6]. The traditional drying process can bring serious nutritional loss, food safety, and cost problems [7,8]. In addition, the traditional drying process impairs the taste and edibility of sea cucumber products. Therefore, the establishment of an effective storage method to maintain the quality of processed sea cucumber is crucial for the development of the sea cucumber industry.

Previous reports showed that sea cucumbers treated by high-pressure steam degraded during storage at 4 °C, which led to the hardness and chewiness dramatically decreasing from 295.85 to 15.67 gf and from 227.08 to 9.18 gf^2^ s, respectively [8]. The hardness and chewiness decreased from 110.53 to 17.93 gf and from 51.58 to 3.82 gf^2^ s, after the sea cucumber was ice-storaged for 8 days [9]. These storage methods are not good enough for sea cucumber quality maintenance. Hence, an improved sea cucumber storage method is necessary to be explored. 

Freezing is considered the easiest way to preserve food for the long term. However, collagen degradation, tissue softening and WHC decrease occur in the sea cucumber during frozen storage [10]. Live sea cucumber storage and transportation are believed to be the most ideal way. At present, some live sea cucumber storage and transportation methods have been developed to overcome autolysis. Sealed polyethylene bags filled with seawater are the most common transportation method. Some studies have found that methods such as adding oxygen can prolong the transportation time of sea cucumbers [11]. Nevertheless, the storage method’s effect on the quality of processed sea cucumber remains unknown. 

This research intends to compare the storage methods’, including the frozen method and keep-alive methods, effect on the quality of processed sea cucumbers. In this study, the quality of the boiled sea cucumber under different techniques of transportation was investigated by the index of water holding capacity (WHC), texture, volatile amino nitrogen (TVB-N), weight loss rate, and Van Gieson (V-G) staining. This research aims to find an effective preservation method to try to avoid impairment of the original taste and texture of sea cucumber. This research would help to further understand the relationship between the storage method and sea cucumber quality change and is beneficial to the sea cucumber industry development.

## 2. Materials and Methods

### 2.1. Materials

Fresh live sea cucumbers (*Apostichopus japonicus*, 100 ± 20 g) in length from 12 cm to 15 cm (shrunken body length) were purchased from the Liujiaqiao Aquatic Products Market in Dalian City. Sea cucumbers were transported to the lab in plastic bags packed in an insulation can without water under the temperature of 0–8 °C within 1 h. Sea cucumbers of 4 treatments were divided into 16 groups for storage tests with each group containing 6 replicates. The extra 6 live sea cucumbers were used as fresh control for the test. All seawater used in this research was prepared from deionized water and the salt of sea crystal (Shanghai Baojia Industry & Trade Co., Ltd., Shanghai, China). A V-G staining solution for V-G staining was purchased from Beijing Regen Biotechnology Co., Ltd. All chemical reagents used in subsequent experiments were analytical grade or above.

### 2.2. The Storage of Sea Cucumbers

After their arrival at the lab, the sea cucumbers were immediately treated. The live sea cucumbers were stored by four methods, including the seawater method, the oxygenation method, the ascorbic acid treatment method, and the frozen method. The live sea cucumbers were distributed in plastic bags, which were 20 cm long × 25 cm height with a total volume of 1 L at the density of 1 individual per bag. The seawater group was filled with 600 mL seawater. The bags of the oxygen group were filled with 600 mL seawater, air, and oxygen (99%, *w*/*w*). The volume of added oxygen accounted for 25% (*v/v*) of the total gas. In the ascorbic acid treatment group, 400 mL of air and 600 mL of seawater were filled into the bags. The ascorbic acid (Soleibao Technology Co., Ltd., Beijing, China) was subsequently added to the seawater to the final concentration of 25 mg/L. The bags were sealed and stored in three refrigerators (KE40, Tulian, Shenzhen, Guangdong, China) for 7 days in darkness at 4 °C. In the freezing group, the sea cucumbers were directly stored in the −20 °C refrigerator. The 4 °C refrigerators were shaken twice a day with 10 min per shake to simulate the transportation conditions. Samples were taken out on days 0, 1, 3, 5, and 7 to test. The sea cucumbers in different treatment groups were washed, the viscera removed, boiled for 5 min at 100 °C, and cooled down to room temperature for tests. Sea cucumbers at day 0 were used as the fresh control. 

### 2.3. Sensory Score Evaluation

After storage and boiling, the external status of the sea cucumbers was classified and evaluated according to the terms in Table 1. The sensory score regulation was according to the standard of Ready-to-Eat Sea Cucumbers (SC/T 3308-2014) [12]. 

### 2.4. Weight Loss Rate Determination

Cooking loss was determined according to Hong et al. [13]. Sea cucumbers were weighed before and after boiling to measure the cooking loss using the following equation:$$\mathrm{Cooking}\;\mathrm{loss}\;(\%)=\frac{m0-m1}{m1}\;\;\times \;100$$
where *m*_0_: the weight of sea cucumber before boiling, g; *m*_1_: The boiled sea cucumber weight, g.

### 2.5. Microstructure of Boiled Sea Cucumber

The boiled sea cucumber body wall was cut to a size of 0.5 cm × 0.5 cm × 0.5 cm, and placed in a fixative containing 10% formaldehyde at 4 °C for 18 h. The specimens were then dehydrated in xylene (20 min-xylene, 20 min-xylene, 5 min-anhydrous ethanol, 5 min-anhydrous ethanol, 5 min-75% alcohol) and washed with tap water. The above procedure was realized through an automatic dehydration machine (Leica TP 1020, Germany) running according to an artificially set program. The samples were dehydrated by a graded ethanol series and embedded in paraffin. Next, the boiled tissues were cut into slices 20 μm thick with a paraffin slicer (Leica RM 2016, Germany). The tissue slice was stained by being subjected to V-G staining (mixing saturated picric acid and acidic magenta 9:1 to form VG dye) for 1 min followed by washing with tap water, and rapid dehydration of anhydrous ethanol. Then, the slices were put into a xylene transparent, neutral gum sealing (transparent sealing: two cylinders of clean xylene transparent for 20 s and 5 min), and neutral gum wet sealing. The microstructures were imaged using a NIKON ECLIPSE E100 optical microscope (NIKON ECLIPSE E100, Japan).

### 2.6. Determination of Water Holding Capacity (WHC)

WHC was determined according to Shi et al. [14] with some modifications. The sea cucumber sample (2 g) was placed in a 10 mL centrifuge tube containing tissue paper to absorb water to centrifuge at 4 °C for 15 min at 10,000× *g*. The released body wall water was measured. In addition, a 2 g sample was re-weighed and placed in an oven at 105 °C for 14 h, and the initial water content was measured by a gravimetric method. Each test had 3 parallels. The WHC was calculated using the following equation:WHC % = (the initial water content − the released water content) × 100/the initial water content 

### 2.7. Texture Properties Analysis

The sea cucumber body wall was cut into 1 cm × 1 cm × 1 cm cubes. Textural parameters including hardness and springiness were measured by a TA-XT2i texture analyzer (Stable Micro Systems, Surrey, UK) with a TPA P/50 square probe. Compression was performed at the rate of 1 mm/s and the interval between two consecutive test cycles was 5 s to compress to 30% deformation. Each test had 6 parallels.

### 2.8. Determination of TVB-N

The volatile base nitrogen content determination was according to the Chinese National Food Safety standard GB5009.228-2016 [15]. The sea cucumber was cut into pieces with surgical scissors and crushed in a meat grinder with a rated power of 37 W for 3 min. Deionized water of 50 mL was added to the crushed sea cucumber sample and left on ice for 30 min. Then, the centrifugation of 5000× *g* was performed at 4 °C for 15 min. The 10 mL supernatant and 1 g magnesium oxide were mixed and put into a digestive tube, then immediately subjected to an automatic Kjeldahl apparatus instrument for determination (VELP UDK+129, Italy). The digestive tube was distilled for 4 min and the released gas was absorbed with a 20 g/L boric acid solution. The absorption solution was calibrated with 0.1 mol/L hydrochloric acid. Distilled water was used as the blank control. TVB-N content was calculated and expressed with a unit of mg/100 g. The calculation formula is as follows:

X = ((V1−V2)×C×14×100)/mV3;

X—Content of volatile base nitrogen in the sample, mg/100 g;

V1—Volume of hydrochloric acid standard titrant consumed, mL;

V2—Volume of standard hydrochloric acid titrant consumed by blank sample, mL;

V3—Measure the volume of sample liquid, mL;

C—The concentration of hydrochloric acid standard titrant, mol/L;

m—Sample weight, g;

14—Titration 1.0 mL hydrochloric acid (1.000 mol/L) standard titration liquid when the mass of nitrogen, g/mol;

100—The calculated results were converted to a conversion factor of mg/100 g.

### 2.9. Protein Degradation Analysis

SDS-PAGE was carried out using the method described by M.D. Suárez [16]. SDS-PAGE with a concentration of 5% stacking gel and 8% separated gel was used for protein composition analysis. The sample was ground on ice with protein extract reagents (20 mM Tris-HCl pH 7.5, 150 mM sodium chloride, 2 mM EDTA, 10 mM DTT, 1% Trition X-100, 0.1% SDS) at a ratio of 1:3 (*w*/*v*). The extract was centrifuged at 12,000× *g* for 15 min at 4 °C, and the supernatant was mixed with a SDS-PAGE sample buffer (0.25 M Tris-HCl pH7.5, 8 M Urea, 5% SDS, 5% mercaptoethanol) in a 4:1 ratio, then boiled for 5 min. The analysis protein concentration was 1.5 µg/ µL and the loading volume was 10 µL. Electrophoresis was performed at a constant voltage of 80 V voltage until the bromophenol blue reached the bottom of the gel. Protein bands were visualized by using 0.05% Coomassie Brilliant Blue R-250 in 45% methanol and 10% acetic acid. 

### 2.10. Statistical Data Analysis

All the tests contained at least three replicates. The IBM SPSS software 19.0 (SPSS Inc. Chicago, IL, USA) was used to conduct a one-factor analysis of variance (ANOVA). Data are reported as mean ± standard deviation (SD). Multiple comparisons between groups were performed by the Duncan method. Comparisons that yielded *p* < 0.05 were considered significant. The experimental images were processed using Image J (National Institutes of Health, Bethesda, MD, USA). All the diagrams were plotted using Origin 8.5 (Origin Lab Corporation, Northampton, MA, USA). 

## 3. Results

### 3.1. Sensory Evaluation of Boiled Sea Cucumber

After seven days of storage and being boiled, the extrinsic feature changes of the sea cucumbers were observed (Figure 1). The sea cucumbers of the oxygen group showed gray to brown with uniform color. The shrinkage of sea cucumber in the frozen group was clearly observed. With the storage time extension, sensory scores of sea cucumbers in all four treatment groups showed a downward trend (Figure 2). Compared with the fresh group, the sensory scores of sea cucumbers in the frozen group and air group significantly decreased on the first day (*p* < 0.05). After 7 days of storage, the highest sensory score was 69.56 ± 3.15 in the oxygen group and the lowest score was 57.67 ± 1.15 in the freezing group.

### 3.2. Histological Changes of Boiled Sea Cucumber

The sea cucumber body wall in different treatment groups were boiled, sliced, stained, and observed to compare the storage effect on tissue structure (Figure 3). The myofibril showed yellow, and the collagen was red after the body wall tissue staining by V-G dye. In the fresh samples, the collagen fibers are intact with regular and compact alignment. After seven days of storage, the space between collagen fibers in the oxygenated group was the smallest and the texture was the best among all treatments. However, the gap between collagen fibers in the frozen group was large and the condition was the worst of all. In the freezing process, the free water in the body wall of the sea cucumber crystallizes, causing the fiber fracture and a gap large between the fibers [17].

### 3.3. Changes in Textural Properties

The texture is one of the most determinant factors that influence the tissue characteristics and consumers’ acceptance of sea cucumber [18]. The hardness, chewiness, springiness, and adhesiveness of storage sea cucumbers were negatively correlated with longer storage time, and the hardness and chewiness significantly decreased (Figure 4A,B). Hardness is an important texture characteristic that determines the quality of sea cucumber. After storage for 1 day, the hardness of the boiled sea cucumbers in different treatment groups was significantly decreased (*p* < 0.05). On day 7, the hardness of the seawater group, oxygen group, ascorbic acid group, and frozen group were 32.0%, 32.52%, 26.95%, and 35.12% of the fresh group (239.66 ± 3.51 N), respectively. The chewiness is the result of a combination of muscle stiffness, cohesion between muscle cells, and reduced muscle elasticity. The chewiness changes of the boiled sea cucumber during seven days of storage are shown in Figure 4B. The chewiness of the boiled sea cucumber on day 0 was 164.72 ± 9.03 N. By day 7, chewiness in the oxygen group (78.98 ± 5.10 N) was higher than in the seawater, ascorbic acid, and frozen groups. This is related to the solubility of collagen in the body wall of sea cucumbers in different treatment groups during storage [19]. Springiness refers to the ability to recover from deformation. After storage for 7 days, the springiness change of the boiled sea cucumbers is shown in Figure 4C. Overall, the springiness of the boiled sea cucumbers in all four groups decreased compared with the initial springiness value of 1.02 ± 0.05. The frozen group showed the fiercest elastic change to 0.65 ± 0.00 after 7 days’ storage. With the extension of storage time, the adhesiveness of the boiled sea cucumber decreased to varying degrees, but the change was not significant (Figure 4D). The seawater group, oxygen group, ascorbic acid group, and freezing group decreased to 0.83 ± 0.00, 0.84 ± 0.00, 0.83 ± 0.02, and 0.86 ± 0.00, respectively, from the initial of 0.91 ± 0.00. Food texture is a major factor in the sensory evaluation of food quality, and it is acted upon by force or deformation in a complex form to cause changes or breakdowns in the structure of the food [20]. Thus, the decline of sea cucumber textural parameters after storage indicated structural damage during storage. The decrease in the four texture parameters is substantially related to the tissue protein degradation and the change in water-holding capability. The deterioration of textural quality might be due to the autolysis occurrence during sea cucumber storage. In addition, the closer to the fresh sea cucumber texture parameters, the better the storage method. Hence, oxygen treatment is the best method to store sea cucumber and is beneficial for maintaining sea cucumber quality. 

### 3.4. Changes in Weightlessness Rate and WHC

Weightlessness rate refers to the ratio of sea cucumber weight before and after boiling. The weightlessness of the sea cucumbers in different treatment groups after boiling are shown in Figure 5A. The weightlessness of the four treatment groups greatly polarized after 7 days of storage. Among these, the oxygen group kept steady weightlessness with a little change of 4.24% ± 0.45% compared with day 0. The weightlessness in the seawater group and ascorbic acid group decreased from day 0 by 78.56% ± 0.90% to 63.37% ± 1.58%, and 62.08% ± 1.11% on day 7, respectively. However, the weightlessness rate of the frozen sea cucumber after boiling increased from 78.56% ± 0.90% to 86.96% ± 0.83%.

Water holding capacity (WHC) is a useful tool for describing protein quality [21]. It is defined as the ability of meat to hold all or part of water (stationary water) during the application of a force [22]. The WHC in the seawater group and oxygen group decreased from 95.33% ± 1.27% to 79.04% ± 1.68% and 81.47% ± 1.97%, respectively (Figure 5B). WHC of the ascorbic acid group showed a decreasing trend from day 0 to day 5 but significantly increased from day 5 to day 7 (*p* < 0.05). In the freezing group, WHC significantly decreased on day 1 and then slowly increased to 93.29% ± 0.32% on day 7.

### 3.5. Changes in TVB-N

The TVB-N content of the sea cucumbers was measured after storage and boiling (Figure 6). The TVB-N content in the seawater group, oxygen group, and ascorbic acid group all showed an upward trend. Among the three groups, the oxygen group showed the smallest increase from day 0 (3.78 ± 0.60 mg 100 g^−1^) to day 7 (10.40 ± 0.12 mg 100 g^−1^). However, the content of TVB-N in the frozen group was the lowest in all the storage groups. This is due to the decomposition-related enzymes in sea cucumbers being inhibited by low temperatures. These enzymes showed no or very weak metabolic capacity, which resulted in less protein decomposition [23]. This result indicated that live sea cucumbers under oxygen intervention showed less deterioration and less protein decomposition among all the live sea cucumber storage treatments, but worse than the frozen method. This is consistent with the pike TVB-N content detection results by Gülsün et al. [24].

### 3.6. Changes in Protein Content/Composition

In order to identify the changes in proteins inside the sea cucumber, the proteins were extracted and analyzed by SDS-PAGE. Three protein bands with molecular weights of 200 kDa (polymer mass, Myosin heavy chain MHC), 44.3 kDa (actin), and 35 kDa (tropomyosin) were used to evaluate the protein degradation during the storage of the sea cucumbers (Figure 7A). The HMW degradation in the seawater group began on day 1 and further intensified from day 3. Quantitative analysis of tropomyosin degradation is shown in Figure 7B (relative expression = tropomyosin/actin). As shown in Figure 7B, the tropomyosin degradation rate in the oxygen group was lower than that in the other three groups in the first five days. However, on day 7, the degradation rates of seawater and oxygen protein were significantly increased (*p* < 0.05). This result indicated that sea cucumber tissue can survive intact for up to five days. On day 7, the sea cucumber protein degradation trend was different from that of the first five days, this is because tissue necrosis occurred on the seventh day of storage.

## 4. Discussion

This study aimed to investigate the effect of different storage methods on the food quality of sea cucumbers. The storage methods included the storage of live sea cucumbers (seawater group, oxygen group, and ascorbic acid group) and the storage of dead sea cucumbers (frozen group). This study identified the sea cucumber deteriorative procedure during storage and demonstrated the best method for sea cucumber storage. 

It is hard to judge the sea cucumber food condition by one indicator, thus, the sensory score became the most dependable analysis method. After 7 days of storage, the sensory score of all live sea cucumbers was significantly higher than that of frozen sea cucumbers. Frozen storage is the simplest method for sea cucumber storage, but the low temperature weakened the meat quality. The frozen group of sea cucumbers showed obvious enhancement of weightlessness rate and WHC compared with the other groups, which indicated that the tissue protein was denatured by low temperature. The denature of tissue protein led to the exposure of the hydrophilic group on the surface of the protein and made it more absorptive to the water molecules [10,25]. The denature of tissue protein in the frozen group of sea cucumbers was also confirmed by the V-G staining result. The histological staining result showed that more myofibrillar was visible in the frozen group than in the other storage groups. The denaturing of the myofibrillar makes it easily stained by staining material. The tissue protein denaturation in frozen sea cucumbers caused the meat’s textural properties to significantly differ from the live storage individuals. Although the TVB-N content is low, the frozen sea cucumber cannot be accepted by sensory score evaluation. 

By contrast, the storage of live sea cucumber resulted in a better food effect. However, live sea cucumbers are prone to autolysis during storage and transportation, which seriously inhibits the food organoleptic quality. The skin of sea cucumbers would show obvious mucous, and the muscle tissue is softened during storage. Therefore, preventing autolysis is of great significance to the quality of sea cucumbers. The oxygen and ascorbic acid treatments are all effective methods to keep sea cucumbers alive during transportation. The sensory score of the oxygen treatment group was the highest in the storage of live sea cucumbers, indicating that the sea cucumbers maintained a good condition and the autolysis occurrence slowed down. 

Borwankar defined texture characteristics as the deformation characteristics of viscoelastic materials under the action of force, expressed as hardness, chewiness, elasticity, viscoelasticity, and other mechanical indicators, and described as the histological characteristics of food [26]. During the whole storage process, the texture properties (hardness, chewiness, elasticity, and adhesion) of sea cucumbers in all groups showed a decreasing trend. About 70% of the total protein in sea cucumber is collagen [27]. Collagen is the main component in the body wall of sea cucumbers, and the texture change is deeply related to it. Hardness is the force required to direct food to achieve a certain deformation, that is, the internal binding force of food to maintain its shape. The chewiness is associated with elasticity and adhesion. Chewiness is equivalent to bite strength, which is the result of muscle hardness, cohesion between muscle cells, and reduced muscle elasticity [28,29]. The decline in all the above factors indicated that the sea cucumber tissue collagen degraded to some extent [8]. Similar results were also reported in squid muscle [30] and cod (*Gadus morhua*) [31]. The tissue staining result also confirmed this. After storage for 7 days and boiling, all group samples exhibited obvious hollow space in the tissue slice, yet no space was found in the fresh group. The more serious the degradation of sea cucumber protein, the larger the network gap structure between collagen fibers will become. The difference in tissue sections in each treatment group is often related to the degradation of structural proteins. Therefore, we also analyzed the degradation of proteins during storage. The oxygen group showed the least protein degradation within 5 days, confirming the intact state of structural protein. Protein degradation also leads to a significant decline in water retention and texture characteristics of sea cucumbers [25].

In the texture properties test, the decline of the oxygen group was the mildest one, demonstrating that oxygenation makes the sea cucumber live in the best condition and postponed the degradation of collagen. The tissue slice of the oxygen group showed the most uniform texture and was very close to the fresh group. In addition, the oxygen group showed the minimum weightlessness rate change. The weight loss rate is the ratio of a sea cucumber’s weight change before and after boiling. The weight in the oxygen group almost did not change before and after boiling, which means that the oxygen storage sea cucumber tissue maintained a healthy level. The water-holding capacity of sea cucumbers affects the sensory characteristics and edible quality of sea cucumbers [9]. It is defined as the ability of meat to hold all or part of water (stationary water) during the application of force. The water-holding capacity of the oxygen treatment group was stronger than that of the seawater group. This is mainly because sea cucumbers in the oxygen group have good health status and less degeneration of collagen in the body wall, which can adsorb water by hydrogen bonds [32]. 

The content of total volatile base nitrogen can be used to judge the freshness of animal food [33]. A high TVB-N value means that fish severely deteriorates [34]. A TVB-N value of 35 mg/100 g is considered the upper limit for acceptability of the fishery products, beyond this limit, the product is considered to be spoilt [35,36]. In the present study, TVB-N levels increased to the top of 13.405 ± 0.917 mg/100 g at day 7 and remained within the acceptable ranges of ≤ 30 mg/100 g. The TVB-N increase was also observed in other aquatic products, such as bighead carp (*Aristichthys nobilis*) fillets [37], rainbow trout [38], and grass carp [39]. During the degradation of sin croaker fish protein, ammonia, dimethylamine, and trimethylamine were produced, which increased the content of TVB-N [21]. In the live sea cucumber storage, compared with the seawater group and ascorbic acid group, the TVB-N value of the oxygen group was the smallest, indicating that the oxygen treatment group produced less ammonia from the degradation of body wall protein and maintained a better tissue state. Thus, oxygen treatment is the best storage method among all tests to maintain the quality of processed sea cucumbers. 

## 5. Conclusions

Frozen storage damaged the tissue myofibrillar protein in sea cucumbers and made the meat texture spoil. Although the TVB-N content was very low, frozen storage is not a good method for sea cucumber storage. By contrast, live sea cucumber storage achieved better conditions, including appearance, sensory score, tissue structure, and textural properties for processed sea cucumber. Despite this, the oxygen storage method showed the best condition in the water-holding ability, texture properties, and tissue microstructure. Oxygen storage is an effective method for sea cucumber storage and gives the best sensory score after storage and boiling. However, further research is essential to be carried out in the future to reveal the mechanism of oxygen storage effect on the sea cucumber tissue change and prevent protein degradation. 

## Figures and Tables

**Figure 1 foods-11-04098-f001:**
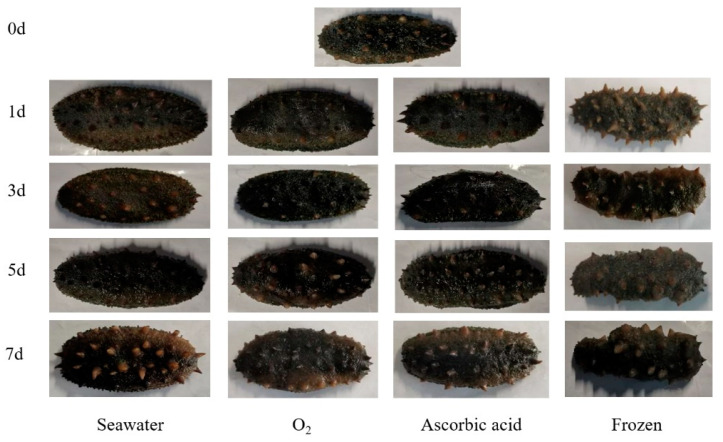
Exterior change of boiled sea cucumbers in different treatment groups during storage.

**Figure 2 foods-11-04098-f002:**
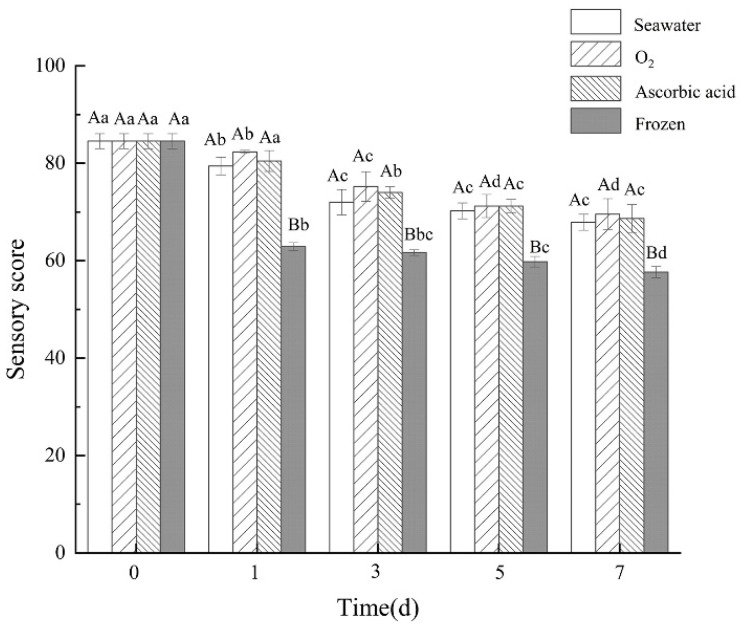
Sensory score of boiled sea cucumbers in different treatment groups during storage. Dates are presented as mean ± SD (*n* = 6). Different capital letters indicate significant differences in the average between groups. Different lowercase letters indicate significant differences in the average value within each group (*p* < 0.05).

**Figure 3 foods-11-04098-f003:**
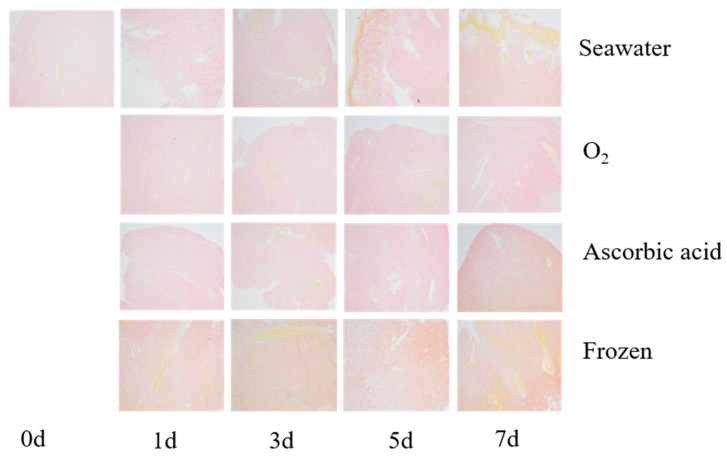
The tissue slice of boiled sea cucumber after storage with different methods for 7 days. The slice was stained by V-G staining (100×).

**Figure 4 foods-11-04098-f004:**
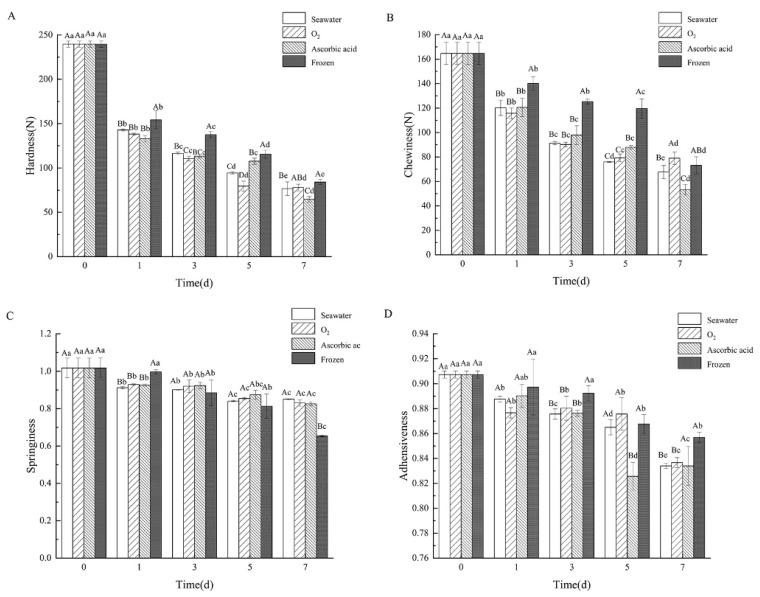
Changes of the (**A**) hardness, (**B**) chewiness, (**C**) springiness, and (**D**) adhesiveness of boiled sea cucumber body wall (SCBW) in different treatment groups during storage. Dates are presented as mean ± SD (*n* = 6). Different capital letters indicate significant differences in the average between groups. Different lowercase letters indicate significant differences in the average value within each group (*p* < 0.05).

**Figure 5 foods-11-04098-f005:**
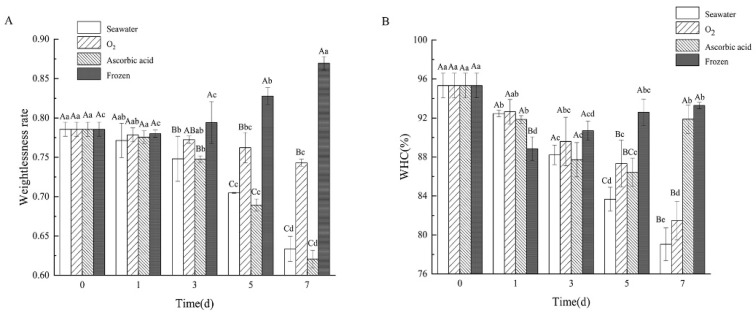
Changes of (**A**) weightlessness rate and (**B**) WHC of boiled SJBW in different treatment groups during storage. Dates are presented as mean ± SD (*n* = 3). Different capital letters indicate significant differences in the average between groups. Different lowercase letters indicate significant differences in the average value within each group (*p* < 0.05).

**Figure 6 foods-11-04098-f006:**
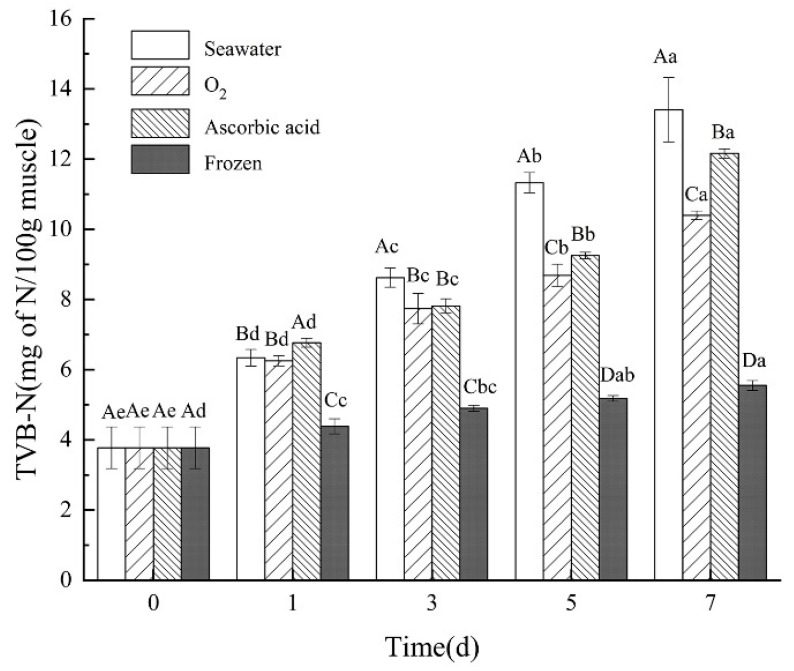
Changes in TVB-N in different treatment groups during storage. Dates are presented as mean ± SD (*n* = 6). Different capital letters indicate significant differences in the average between groups. Different lowercase letters indicate significant differences in the average value within each group (*p* < 0.05).

**Figure 7 foods-11-04098-f007:**
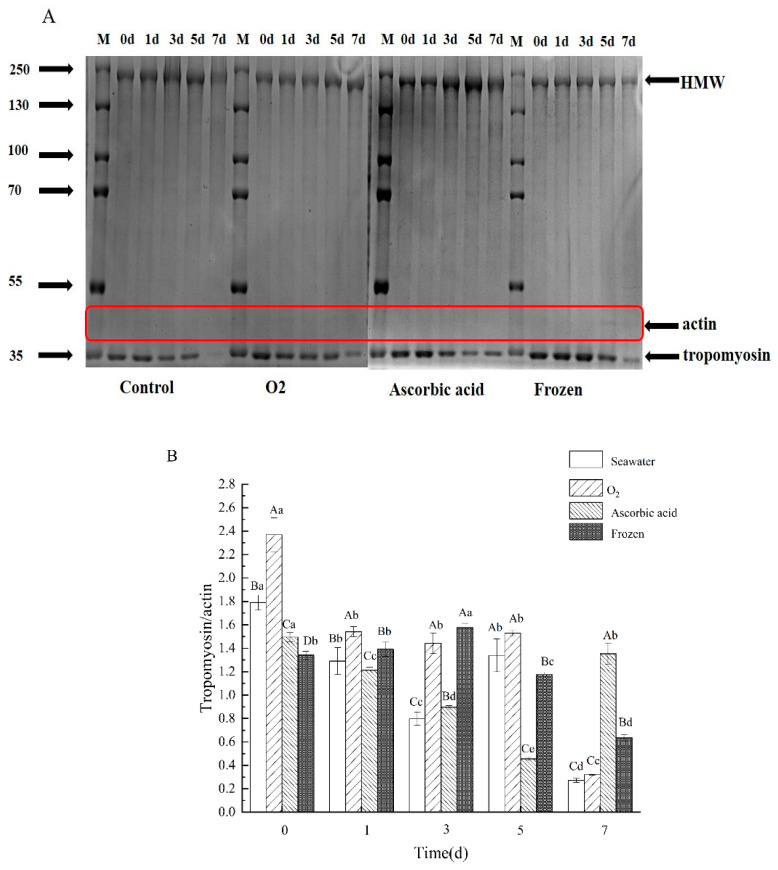
SDS-PAGE (5% Stacking gel and 8% separated gel) analysis of stored sea cucumbers protein by different treatments. (**A**) Electrophoretic diagram of SDS-PAGE and (**B**) quantitative analysis of electrophoresis. Relative expression = tropomyosin/ actin. Dates are presented as mean ± SD (*n* = 3). Different capital letters indicate significant differences in the average between groups. Different lowercase letters indicate significant differences in the average value within each group (*p* < 0.05).

**Table 1 foods-11-04098-t001:** Sensory score evaluation of stored and boiled sea cucumbers.

External Indicators (Score)	Level 1 (20–25 Points)	Level 2 (14–19 Points)	Level 3 (8–13 Points)	Level 4 (1–7 Points)
Color (25 points)	Gray to brown, color is more uniform	Slightly lighter in color	The color becomes lighter and more uneven	Faded serious, uneven color
Organization status (25 points)	Fleshy tissue normal, fleshy elastic	The meat is slightly loose, the elasticity is slightly poor	Meat slack, elasticity is poor	The meat is very hard, the elasticity is very poor
Smell (25 points)	The unique smell of sea cucumber, no peculiar smell	With a slightly fishy smell	The heavier fishy smell	Stench
Impurity (25 points)	No visible foreign impurities	A little impurity	More impurities	A lot of impurities

## Data Availability

Data is contained within the article.

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
