# Peer review of "Effects of Storage Method on the Quality of Processed Sea Cucumbers (Apostichopus japonicus)"

_foods, 2022, doi:10.3390/foods11244098_

Round 1
Reviewer 1 Report
Thanks for good work. Comments are in the attached file.

Reviewer 2 Report
The manuscript contains interesting data. The results are quite well presented and discussed. However, there are some detailed comments:
Please add the numerical data to the Abstract.
line 28: Please add more recent data.
The Introduction does not provide sufficient background. It does not include all relevant references. Therefore, the Introduction should be supplemented.
The novelty of the research is not adequately indicated on the background of available literature.
Subsection 2.1. Materials should be more detailed.
The number of repetitions should be provided for each type of measurement.
2.10. Statistical data analysis: Were the assumptions for the analysis of variance tested?
The directions for further research should be indicated.
Round 2
Reviewer 2 Report
The authors considered each comment.